

# Helium in the Earth's foreshock: a global Vlasiator survey

Markus Battarbee[1], Xochitl Blanco-Cano[2], Lucile Turc[1], Primož Kajdič[2], Andreas Johlander[1], Vertti Tarvus[1], Stehphen Fuselier[3,4], Karlheinz Trattner[5], Markku Alho[1], Thiago Brito[1], Urs Ganse[1], Yann Pfau-Kempf[1], Mojtaba Akhavan-Tafti[6,7], Tomas Karlsson[8], Savvas Raptis[8], Maxime Dubart[1], Maxime Grandin[1], Jonas Suni[1], and Minna Palmroth[1,9]

[1]Department of Physics, University of Helsinki, Finland
[2]Instituto de Geofisica, Universidad Nacional Autonoma de Mexico, Mexico City, Mexico
[3]Southwest Research Institute, San Antonio, Texas, USA
[4]University of Texas San Antonio, San Antonio, USA
[5]University of Colorado Boulder, Boulder, Colorado, USA
[6]Climate and Space Sciences and Engineering, University of Michigan, Ann Arbor, MI, USA
[7]LPP, CNRS, École Polytechnique, Sorbonne Université, Université Paris-Saclay, Observatoire de Paris, PSL Res Université, Institut Polytechnique de Paris, Palaiseau, France
[8]School of Electrical Engineering and Computer Science, KTH Royal Institute of Technology, Stockholm, Sweden
[9]Finnish Meteorological Institute, Helsinki, Finland

**Correspondence:** Markus Battarbee (markus.battarbee@helsinki.fi)

**Abstract.**

The foreshock is a region of space upstream of the Earth's bow shock extending along the interplanetary magnetic field. It is permeated by shock-reflected ions and electrons, low-frequency waves, and various plasma transients. We investigate the extent of the $He^{2+}$ foreshock using Vlasiator, a global hybrid-Vlasov simulation. We perform the first numerical global survey
of the helium foreshock, and interpret some historical foreshock observations in a global context.

The foreshock edge is populated by both proton and helium field-aligned beams, with the proton foreshock extending slightly further into the solar wind than the helium foreshock, and both extend well beyond the ULF wave foreshock. We compare our simulation results with MMS HPCA measurements, showing how the gradient of suprathermal ion densities at the foreshock crossing can vary between events. Our analysis suggests that the IMF cone angle and the associated shock obliquity gradient
can play a role in explaining this differing behaviour.

We also investigate wave-ion-interactions with wavelet analysis and show that the dynamics and heating of $He^{2+}$ must result from proton-driven ULF waves. Enhancements in ion agyrotropy are found in relation to, e.g., the ion foreshock boundary, the ULF foreshock boundary, and specular reflection of ions at the bow shock. We show that specular reflection can describe many of the foreshock ion VDF enhancements. Wave-wave-interactions deep in the foreshock cause decoherence of wavefronts,
allowing $He^{2+}$ the be scattered less than protons.



# 1   Introduction

The Earth's bow shock forms due to the interaction of the supermagnetosonic solar wind with our planet's magnetic field. As in other heliospheric shocks, solar wind particles interacting with the shock undergo a variety of processes including reflection and acceleration. Upstream of the bow shock, in regions where plasma is magnetically connected to the shock, the reflected particles form a region called the foreshock. It is a very complex environment, populated by a variety of suprathermal ion distributions (Thomsen, 1985; Fuselier, 1995; Wilson III, 2016), waves (Hoppe et al., 1981; Blanco-Cano et al., 2009; Wilson III, 2016) and nonlinear transient structures (Kajdič et al., 2017; Blanco-Cano et al., 2018). The edges of the foreshock are magnetically connected to quasi-perpendicular regions of the Earth's bow shock (where the angle between the shock normal and the magnetic field $\theta_{Bn} \gtrsim 45°$) whereas the central region of the foreshock is magnetically connected to the quasi-parallel bow shock (where $\theta_{Bn} \lesssim 45°$).

Most studies of the foreshock have concentrated on studying proton dynamics and properties of ultra-low frequency (ULF) waves. Suprathermal ion distributions in the foreshock include field-aligned ion beams (FABs), gyrating distributions and hot diffuse populations. The original classification based on 2D ISEE (International Sun-Earth Explorer 1) velocity distributions also included intermediate ions (Thomsen, 1985). Subsequent observations with higher time resolution have however showed that intermediate distributions often display signatures of gyrating ions, which can be either isotropic or gyrophase-bunched (Fuselier et al., 1986; Meziane et al., 2001). The interaction of suprathermal ions with the solar wind results in instabilities able to generate ULF waves (Gary, 1991).

Little attention has been given to the helium component, which is the most important minor species in the solar wind. Although helium constitutes typically only about $4-5\%$ of the total ion number density (Ipavich et al., 1984; Wurz, 2005; Gedalin, 2017), its contribution to the upstream mass density and dynamical pressure can be as large as $20\%$ (Gedalin, 2017). Thus, $He^{2+}$ effects in shock dynamics and foreshock physics should not be ignored. Scholer et al. (1981) reported on ISEE observations of proton and alpha-particle $30-36\,keV\,q^{-1}$ beams at the edge of the foreshock, exhibiting similar time profiles. Ipavich et al. (1988) studied the content of $\sim 10$ keV/nuc $H^+$ and $He^{2+}$ in field-aligned beams with the AMPTE CCE spacecraft. They found that the alpha-particles in the beams have approximately the same velocity as the $H^+$ ions, but that the $He^{2+}$ to $H^+$ density ratio is dramatically smaller (two orders of magnitude) than that measured simultaneously in the solar wind. Based on a study of 14 field-aligned beam events recorded with the ISEE satellite, Fuselier and Thomsen (1992) concluded that the ratio is roughly 1/10 of the solar wind ratio.

Fuselier et al. (1990) showed that two types of suprathermal $He^{2+}$ distributions can be observed upstream of the quasi-parallel shock: A diffuse (energetic, from several $keV/e$ up to the detector maximum) distribution, and a nongyrotropic gyrating distribution. These gyrating $He^{2+}$ distributions are observed near the shock and their velocity components are consistent with near-specular reflection of a portion of the incident solar wind $He^{2+}$ ions. The helium content in these gyrating populations can be roughly the same as in the pristine solar wind when the Alfvénic Mach number is $M_A > 7$. These authors suggested at that time that the near-specularly reflected $He^{2+}$ ions may be the seed population for the more energetic diffuse helium populations.





Using ISEE data, Fuselier et al. (1995) and Fuselier (1995) studied in more detail the origin of diffuse suprathermal ions with energies from a few up to $\sim$100 keV/$e$. They found that the suprathermal $He^{2+}$ fraction (normalized to solar wind abundances) is dependent on location within the foreshock. High-energy field-aligned beams ($> 10$ keV/$e$) observed near the foreshock edge contain a significant fraction of $He^{2+}$ (near solar wind quantities), whereas lower-energy beams ($\sim$1 keV/$e$) found deeper in the foreshock and in association with intermediate proton distributions, have low helium fractions. This difference in helium

fraction was then assumed to be indicative of their origin. Low energy beam production was explained in terms of magnetospheric leakage or shock reflection, whereas high energy beams were attributed to shock drift acceleration, which is efficient for both protons and $He^{2+}$. Additionally, Fuselier et al. (1995) found that high fraction (similar to solar wind concentration) suprathermal $He^{2+}$ are seen only in the quasi-parallel foreshock, their distributions are nearly always nongyrotropic partial ring beams, and are consistent with specular reflection of a portion of the incident solar wind (Fuselier et al., 1990). These

$He^{2+}$ ions are found in association with ring beam $H^+$ distributions, also associated with specular reflection.

  Diffuse ions in the foreshock usually have the same concentration of helium as the solar wind $n_{\mathrm{p}}/n_\alpha \sim 4\%$. Thus, Fuselier et al. (1995) suggested that the lower energy field-aligned beams with almost no helium content cannot be the seed population for diffuse ions. They proposed that the very energetic field-aligned beams at the edge of the foreshock propagate upstream much faster than the solar wind flow and are confined to the edge, and therefore cannot contribute to the diffuse population

observed further downstream. These results changed the original paradigm where the origin of diffuse ions was explained in terms of field-aligned beams evolving into intermediate ions and then diffuse distributions (Thomsen, 1985). The fact that high concentration $He^{2+}$ gyrating ions are observed in the quasi-parallel foreshock, as are diffuse ions, suggests that gyrating distributions can be the seed population for the diffuse $He^{2+}$ distributions. Similarly, gyrating $H^+$ distributions are probably the source of the energetic diffuse $H^+$ as well.

Using 1-D hybrid simulations, Trattner and Scholer (1991) and Trattner and Scholer (1994) studied the acceleration of protons and $He^{2+}$ ions at the quasi-parallel shock. They found that the concentration of helium in the diffuse population depends on the solar wind Mach number, plasma $\beta$, and the shock $\theta_{Bn}$. In another numerical work, Trattner and Scholer (1993) investigated the thermalization of $He^{2+}$ through the quasi-parallel shock, and showed that even if initially the heavier ions are less decelerated by the cross-shock potential, this difference disappears within a few gyroperiods downstream of the

shock. The simulation results of Trattner and Scholer (1994) show a nongyrotropic $He^{2+}$ distribution re-entering the upstream region from the magnetosheath due to its large gyroradius, consistent with the shapes of the distributions observed by ISEE. These authors showed that $He^{2+}$ ions can alter the shock structure and that the occurrence of $He^{2+}$ ion clouds upstream of the shock is dependent on Mach number.

  In the recent years, the Hot Plasma Composition Analyzer (HPCA) instrument (Young et al., 2016) onboard the Magne-

80 tosphere Multiscale (MMS) spacecraft (Burch et al., 2016) has allowed new investigations of helium ions near the Earth's bow shock, providing in particular $He^{2+}$ velocity distributions. Broll et al. (2018) investigated the reflection of $He^{2+}$ at the quasi-perpendicular bow shock and showed that $He^{2+}$ ions can undergo a similar specular reflection process as the protons. The study of interstellar $He^+$ pick-up ions at the quasi-perpendicular bow shock conducted by (Starkey et al., 2019) revealed that single reflection at the shock plays a significant role in accelerating these ions.





In this work, we analyze $He^{2+}$ properties in the foreshock using a global hybrid-Vlasov simulation of near-Earth space performed with the Vlasiator model (Palmroth et al., 2018). Vlasiator ion distribution functions have been compared to spacecraft observations of the foreshock in the past in Kempf et al. (2015). We investigate both the local and global properties of suprathermal $He^{2+}$ ions and their possible influence on wave activity. We compare our results with MMS measurements in the Earth's foreshock, and propose some new interpretations of some ISEE observations in the global context provided by our

numerical simulation.

## 2   Methods and data

### 2.1   Simulations

We investigate the Earth's foreshock region using Vlasiator (Palmroth et al., 2018), a hybrid-Vlasov simulation capable of describing ion kinetics whilst encompassing global scales. Vlasiator solves the Vlasov equation for grid-discretized particle

distribution functions, with closure provided by Ohm's law augmented by the Hall term. Electrons are considered a charge-neutralizing massless fluid. We investigate the foreshock using a 2D-3V-simulation, with 3D moments and velocity distribution functions but a 2D spatial domain. The Geocentric Solar Ecliptic (GSE) simulation domain is $X \in [-48.66\,R_\mathrm{E}, 64.35\,R_\mathrm{E}]$ and $Z \in [-59.65\,R_\mathrm{E}, 39.24\,R_\mathrm{E}]$ and a single cell width in the $Y$-direction. The spatial resolution is $300\,\mathrm{km}$ (1.3 times the solar wind ion inertial length) and the velocity space resolution for both ion species (protons and alpha-particles) is $30\,\mathrm{km\,s^{-1}}$.

Our simulation is set to have solar wind values of $\beta = 0.7$, $\mathrm{M_{ms}} = 5.6$, and $\mathrm{M_A} = 6.9$. We initialize the simulation with a solar wind of $n_\mathrm{p} = 1\,\mathrm{cm^{-3}}$ and $n_\alpha = 10^{-2}\,\mathrm{cm^{-3}}$. Due to the mass ratio, we set $T_\mathrm{p} = 0.5\,\mathrm{MK}$ and $T_\alpha = 1.0\,\mathrm{MK}$. The solar wind speed is set to $u_\mathrm{sw} = 750\,\mathrm{km\,s^{-1}}$ in the $-\hat{e}_x$ direction, simulating fast solar wind conditions and ensuring efficient simulation initialization. We set an IMF of $B_x = 3.54\,\mathrm{nT}$ and $B_z = -3.54\,\mathrm{nT}$ resulting in a $45°$ cone angle. Although the simulation plane is meridional, the foreshock dynamics are comparable with an ecliptical Parker spiral set-up. The Earth's magnetic dipole is

a $\hat{e}_z$-aligned line dipole resulting in a realistic magnetopause standoff distance (Daldorff et al., 2014). The simulation has an inner boundary at $3 \cdot 10^4\,\mathrm{km} \approx 4.7\,R_\mathrm{E}$, modeled as a perfectly conducting ionosphere. Thus, the run is nearly identical to the simulation presented in Blanco-Cano et al. (2018), with the addition of alpha-particles as an independent, self-consistent species. Our alpha-particle density is set to only 1% of the solar wind so mass loading and effects on ULF wave properties are expected to be small. In order to constrain memory usage, we set the minimum stored phase-space densities (as explained in

von Alfthan et al., 2014) to $f_{\mathrm{min,p}} = 10^{-15}\,\mathrm{s^3\,m^{-6}}$ and $f_{\mathrm{min},\alpha} = 10^{-17}\,\mathrm{s^3\,m^{-6}}$.

     Additionally, in subsection 3.1 we compare our main simulation to an equatorial Vlasiator simulation with a spatial resolution of $228\,\mathrm{km}$ and solar wind values of $\beta = 2.3$, $\mathrm{M_{ms}} = 5.9$, $\mathrm{M_A} = 10$, $n_\mathrm{p} = 3.3\,\mathrm{cm^{-3}}$, $n_\alpha = 3.3 \cdot 10^{-2}\,\mathrm{cm^{-3}}$, and $u_{\mathrm{sw},x} = -600\,\mathrm{km\,s^{-1}}$. The IMF is set to $5\,\mathrm{nT}$ with a $5°$ cone angle.





## 2.2 Observations

In this study, we also analyze observations from the MMS mission (Burch et al., 2016) in the Earth's foreshock. We use data from three different instruments: the fluxgate magnetometer (FGM, Russell et al., 2016), the Hot Plasma Composition Analyzer (HPCA, Young et al., 2016), and the Dual Ion Spectrometers (DIS, Pollock et al., 2016) which is part of the Fast Plasma Investigation (FPI) instrument. For all instruments, we use survey mode data only. The FGM produces magnetic field measurements with $16\,\mathrm{s}^{-1}$ time resolution in fast survey mode.

Energy-time spectrograms from HPCA measurements are used to identify whether the spacecraft are located in the foreshock or in the solar wind. We also calculate partial densities for the different ion species, following the procedure described in the HPCA Science Algorithms and User Manual available on the MMS Science Data Center page[1]. HPCA data are made available in survey or burst modes. Data are collected in 0.5 spin (10 seconds) time resolution at 64 energies, 16 azimuth angles and 16 elevation angles for five different species ($H^+$, $He^+$, $He^{2+}$, $O^+$ and $O^{2+}$). In the survey mode data used here, the data are

reduced in energy and angles to 16 energies, 8 azimuths, and 8 elevation angles. The total energy range of the data is between $\sim 1\,\mathrm{eV}/e$ and $40\,\mathrm{keV}/e$ with a mass resolution of $M/\Delta M \sim 8$.

Finally, we calculate the ion partial density (without species separation) using measurements from FPI-DIS, as a means of comparison to the partial densities obtained from HPCA. The FPI produces burst skymaps which consist of ion count arrays of 32 energies × 32 azimuth angles × 16 polar angles that are accumulated every 150 ms. Then, 30 consecutive DIS burst

skymaps are summed in order to produce the survey mode skymaps with time resolution of 4.5 seconds. The energy range of the DIS is between $10\,\mathrm{eV}/e$ and $30\,\mathrm{keV}/e$.

## 3 Results

In Figure 1, we show an overview of the foreshock region in the simulation. The top panel shows out-of-plane magnetic field $B_y$ fluctuations at time $t = 1100\,\mathrm{s}$, showcasing the ULF wave fronts seen throughout the foreshock, displayed on a symmetric log-

135 arithmic colour scale. The black curves indicate magnetic field lines. The bottom panel also displays the ULF foreshock extent as black contours drawn for $B_y = \pm 0.1\,\mathrm{nT}$ at time $t = 1100\,\mathrm{s}$, with the diverging colourmap indicating the relative abundances of foreshock suprathermal ions, scaled to the incoming solar wind number density ratio. The solar wind / foreshock thermal ion distribution was accumulated from the velocity space constrained within a sphere of $500\,\mathrm{km\,s}^{-1}$, centered at the solar wind speed $u_{\mathrm{sw,x}} = -750\,\mathrm{km\,s}^{-1}$, and all ions outside this sphere were considered part of the suprathermal distribution. Suprather-

140 mal particle measurements were averaged over 4 minutes (between 1080 and 1220 s), providing an overview of a steady state foreshock, smoothing over effects due to the gyration of particle populations at the foreshock edge. An animated version of panel b of Figure 1 showing instantaneous density ratios instead of the 4-minute average is provided as supplementary video A.

---

[1]https://lasp.colorado.edu/mms/sdc/public/datasets/hpca/



### 3.1 Foreshock edge

As shown in Figure 1, the ion and ULF foreshocks are not identical in extent. The ULF foreshock edge visible in both panels connects to the bow shock at $Z \sim -5\,R_{\mathrm{E}}$, whereas the ion foreshock edge intersects the bow shock already at $Z \sim +5\,R_{\mathrm{E}}$ (bottom panel). At the bottom of the figure at $Z = -50\,R_{\mathrm{E}}$ we see the ion foreshock extending up to $X = 40\,R_{\mathrm{E}}$, whereas the ULF foreshock only extends to a position $\sim 10\,R_{\mathrm{E}}$ further downstream. We also see that in panel b) the ratio of suprathermal alphas to protons in the foreshock shows significant deviations from the incoming solar wind ratio of $1/100$. Throughout most of the deep foreshock, the $n_{\alpha,\mathrm{st}} n_{\mathrm{p,st}}^{-1} \cdot 10^2$ shown in Figure 1b ratio tends to values $\gtrsim 2$, whereas at the foreshock edge, it falls below $0.2$. Right at the edge of the foreshock we see spatially periodic structures with again large alphas-to-protons ratios, likely caused by bursty reflection at the quasi-perpendicular shock and the alpha-particles having larger gyroradii, thus gyrating further into the upstream. The ULF and ion dynamics of the foreshock edge are further examined in Figure 2.

The left panel of Figure 2 shows an excerpt from Figure 1a, featuring a portion of the foreshock edge. The colourmap is again the out-of-plane magnetic field component, overlaid with contours of proton (black) and helium (green) suprathermal densities. We focus on the contours upstream of the ULF foreshock. The solid contours are drawn where the suprathermal density is 0.5% of the species' solar wind density, and the dotted contours at 0.1%, respectively. The thick grey lines indicate four cut-throughs across the foreshock edge. The magnetic field strength and the suprathermal proton and helium densities along these cuts are plotted in the right-hand panels of Figure 2. As shown in these plots, there is variation in the profiles of ions across the foreshock edge, moving into the foreshock from right to left. Close to the bow shock (panels b and c) there is a somewhat rapid increase in ion densities near $10\,R_{\mathrm{E}}$ followed by a gradual increase over the following $1-2\,R_{\mathrm{E}}$ and finally plateauing values. Further out (panels d and e) we see a more gradual increase in suprathermal ion densities over several $R_{\mathrm{E}}$ and a less clear plateau. The suprathermal ion density threshold at 0.5% of the species solar wind density is shown as a horizontal dashed line.

We note that non-thermal particles are found several $R_{\mathrm{E}}$ upstream of the foreshock waves, consistent with previous works showing that no ULF wave activity is observed in conjunction with the region closest to the foreshock edge, where field-aligned beams are expected (and found). More importantly, we find significant amounts of suprathermal helium throughout the field-aligned beam region and into the ULF foreshock. As shown by the black and green contours in Figure 2a, the helium foreshock edge is located equal or slightly downstream of the proton foreshock edge. This shift is more pronounced in the dotted contours, and increases when moving further away from the bow shock. At about $30\,R_{\mathrm{E}}$ from the shock, the difference is of the order of one $R_{\mathrm{E}}$. The shift of the foreshock edge is also noticeable in the line profiles (panels b–e). This suggests that the proton foreshock is slightly more extended than the helium foreshock, and measurements made at the very edge of the foreshock can suggest significantly lower helium fractions despite the helium abundances rising to proton-comparable levels a few $R_{\mathrm{E}}$ deeper within the foreshock. We also point out that the plateau of alpha-particles inside the foreshock edge has more fluctuations to it than the plateau of protons.

We also note that the suprathermal ion density contours in Figure 2 as well as the time-averaged ratios seen in Figure 1 are not smooth, instead having a wavy shape. Supplementary Video A shows the wavy shape to be due to intermittent reflection of





ions at the bow shock with bursts of ions propagating away from the shock along the field lines. This periodic and intermittent enhanced reflection of particles may be due to mesoscale reformation of the bow shock (Battarbee et al., 2020e), and can cause

further discrepancies and variation in field-aligned beam densities.

We now compare our numerical results with observations from the MMS spacecraft at the foreshock edge. Figures 3 and 4 show two time intervals during which the MMS3 spacecraft crossed from the solar wind into the foreshock region. The panels from top to bottom show the IMF magnitude and components, the suprathermal densities of $H^+$ (black, HPCA), $He^{2+}$ (green, HPCA) and all ions (red, FPI), and proton and $He^{2+}$ energy-time spectrograms from the HPCA instrument. The $He^{2+}$

suprathermal densities have been multiplied by 10 or 25 to ease comparison of suprathermal ion density gradients. The lowest energy used in the calculation of the suprathermal (partial) densities for each species is also stated on the panel. This energy is about four times higher for alphas compared to the protons due to larger mass of $He^{2+}$. We note that the energy-time spectrograms show energy per charge $E_i Z_i^{-1}$, with the lowest energy included in the suprathermal population shown with the dashed line set to $2.5\,\mathrm{keV}/e$ for protons and $5.0\,\mathrm{keV}/e$ for $He^{2+}$.

During both time intervals the transition between the foreshock and the undisturbed solar wind was not due to any IMF rotational discontinuity, as can be seen from the MMS magnetic field components. We also checked the solar wind measurements propagated to the bow shock nose from the OMNI database (King and Papitashvili, 2005) and found no sharp IMF change during those intervals. This means that the foreshock was undergoing gradual motion due to slow IMF rotations, so the spacecraft did not observe traveling foreshocks (Kajdič et al., 2017) or foreshock cavities (Schwartz et al., 2006; Billingham

et al., 2008, 2011). The spacecraft thus slowly entered or exited the foreshock region so the magnetic field and suprathermal density profiles can be compared to those obtained from our simulations (Figure 2).

Panels 3b) and 4b) show that the proton and $He^{2+}$ suprathermal densities do not always behave in the same manner. In Figure 3b) we can clearly see that the proton and $He^{2+}$ suprathermal densities are well correlated across the foreshock edge, albeit $H^+$ has a slightly stronger initial beam before 22:06. Both increase with a similar relative gradient as the spacecraft

crosses from the solar wind into the foreshock. In Figure 4b the suprathermal proton density starts to increase $\sim 3$ minutes before the suprathermal $He^{2+}$ density and remains low well into the foreshock. This suggests that the proton foreshock extends further outward than the $He^{2+}$ foreshock. In other words, the proton foreshock extends to field lines connected to larger $\theta_{Bn}$ values than the $He^{2+}$ foreshock.

In order to better understand the different foreshock edge crossing behaviour seen in Figures 3 and 4, we compare the main

Vlasiator simulation used in this study to an equatorial plane simulation with a quasi-radial IMF, as detailed in subsection 2.1. In Figure 5 we show a zoom-in to the foreshock edge of both runs, with colourmaps indicating the out-of-plane component of the magnetic field and black and green contours indicating proton and helium suprathermal densities at 0.5% (solid) and 0.1% (dotted) inflow densities, averaged over 2 minutes. In comparing the Vlasiator plots with MMS observations, it is important to keep in mind that the Vlasiator definition for suprathermals includes reflected particles with simulation frame energies

comparable with the solar wind bulk, whereas the MMS suprathermal density is defined with a markedly higher minimum energy threshold. Comparison of the two panels indicates that simulations can reproduce the two different foreshock edge





behaviours, with panel a (our main run) presenting a relatively rapid drop-off, and panel b (the quasi-radial IMF comparison run) showing a much more gradual fall-off of suprathermal ion densities and a greater difference between the two ion species.

The IMF prior to the 30th December 2018 foreshock edge crossing had the IMF pointing in the dawn-antisunward direction
with its cone angle $\sim 55°$. Consequently, this was also the orientation of the foreshock. At the time MMS3 was located near the nose of the Sun-Earth line, at approximately $(12.5, 0.8, 5.1)\,R_{\mathrm{E}}$ in GSE coordinates. This location, together with large IMF cone angle, means that the foreshock edge was crossed in a way qualitatively similar to that shown in the Figure 5a.

In the case of the 18th November 2018 event the IMF cone angle was $\sim 35°$ and pointing in the north-antisunward direction prior to the foreshock encounter, and consequently this was also the foreshock orientation. The GSE coordinates of MMS3
were $(9.8, 11.3, 5.7)\,R_{\mathrm{E}}$ i.e. far from the Sun-Earth line. Thus, this foreshock edge crossing is comparable with the one shown in Figure 5b.

### 3.2   Velocity distribution functions and their properties

We now examine the properties of ion velocity distribution functions (VDFs) at the edge of and within the foreshock. Figure 6 shows 2-D projections of proton and helium velocity distribution functions in the solar wind frame at three positions close
to the foreshock edge, extracted from our Vlasiator simulation at time 1000 s. Subpanels labeled $[\|\perp]$ have been generated by averaging the instantaneous VDFs over the $v_{B \times V}$ direction, whereas subpanels labeled $[\perp\perp]$ have been averaged over the $v_B$ direction. We use two different colour scales to differentiate the phase-space densities of the two ion species, with ranges selected to account for the input solar wind abundance ratio. Panels (1), (2), and (4) show VDFs from virtual spacecraft locations A, B, and C, respectively. The $[\|\perp]$ subpanels also show an ellipsoid located at the position in velocity space where
particles would end up if they were specularly reflected from the solar wind population at the closest point of the bow shock. The shock location was determined according to a plasma compression criterion of $n_{\mathrm{p}} > 2n_{\mathrm{p,sw}}$ and the shock normal direction was estimated to be equal to a vector pointing from the closest shock location to the virtual spacecraft location. We emphasize that this estimate is a rough one, to be improved upon in future studies, and does not account for ion propagation times or drifts. Panel (3) shows an overview of the foreshock region indicating the locations of the spacecraft on top of a colourmap
depicting the temperature anisotropy calculated from the whole proton VDF. The dark orange region at the upstream edge of the foreshock with parallel temperatures in excess of perpendicular temperatures is indicative of the FAB region of the proton foreshock. Magnetic fields lines are depicted with black curves.

In panel (1) of Figure 6, at the position of virtual spacecraft A located very close to the foreshock edge, we see very clear proton and helium FABs, though the helium beam appears to have more structure. Parallel velocities of both beams are
between 1500 and $2000\,\mathrm{km\,s^{-1}}$ in the solar wind frame or $\gtrsim u_{\mathrm{sw}} = 750\,\mathrm{km\,s^{-1}}$ in the simulation frame, which translates to roughly $3 - 5\,\mathrm{keV\,nuc^{-1}}$. We note that this is larger than the energy at which specularly reflected particles would be found, which suggests that these particles have experienced shock drift acceleration (SDA) at the quasi-perpendicular shock front. This indicates that the source of the FABs in panel (3) is located at roughly $X = 16\,R_{\mathrm{E}}, Z = 0\,R_{\mathrm{E}}$. Panel (2) of Figure 6 depicts VDFs at position B, at the boundary between the FAB region and the ULF foreshock. At this location the ion beam
seems to be transitioning into a more gyrating distribution, with a gyrophase-bunched signature (visible in the $[\perp\perp]$ panel at



$v_{B \times V} > 500 \, \mathrm{km\,s^{-1}}$) for both species but in particular for helium at $v_{B \times V} > 1000 \, \mathrm{km\,s^{-1}}$. Parallel velocities have begun to decrease, extending down to $1000 \, \mathrm{km\,s^{-1}}$ in the solar wind frame of reference. A similar upstream ion velocity decrease was also see in Battarbee et al. (2020e). Still, at this position, both helium and protons show a qualitatively similar shape to their distribution functions. Conversely, in panel (4) of Figure 6 at position C, the proton VDF resembles a low-energy field-aligned

beam, extending from $1000$ to $1800 \, \mathrm{km\,s^{-1}}$ in $v_B$, whereas the helium distribution has very little trace of a beam, instead consisting of a broken-up gyrating ion population. An animated version of Figure 6 can be found as Supplementary Video B. We also note that early in this video, at virtual spacecraft A, in particular the helium VDF shows what appears like a gyrophase-bunched population but which remains stationary in velocity space. We deduct that it is in fact spatial sampling of the helium foreshock edge, visible due to the large gyroradius of alpha-particles. We also point out that the proton beam energy found

in particular in panel (1) does not extend to the tens of $\mathrm{keV}/e$ found in some spacecraft observations, possibly a result of the sparse velocity space implementation in Vlasiator.

Figure 7 is similar to Figure 6 but with virtual spacecraft locations chosen to represent regions further within the foreshock and also closer to the quasi-parallel bow shock. Panels (1) and (2) show spacecraft D and E which are located close to the bow shock and depict mostly gyrating and intermediate ion populations. The helium populations of gyrating ions appear to have

260 more structure to them. It is also noteworthy that both proton and helium $[\|\perp]$ subpanels of panel (1) have what looks like a FAB propagating towards the shock at a parallel velocity greater than the solar wind speed, at velocities $v_B < 0$. Although the estimate of the velocity space location of specularly reflected particles does not account for particle travel times or shock shape evolution, we find that the near-circular ellipsoids indicating potential specular reflection are usually found in VDF regions where there are enhancements, suggesting that specular reflection indeed plays a role in the generation of these populations. As

the bow shock reforms as a mesoscale process with distinctly non-planar features, the direction of specular reflection varies, leading to the intermittent and gyrating partial rings seen in these VDFs. This mapping of specular reflection can also be seen in Supplementary Video C which is an animated version of Figure 7. Panel (4) of Figure 7 shows proton and helium VDFs further within the deep foreshock, and away from the bow shock. The proton population appears to be a gyrating ion population and the helium population a low-energy beam population, but viewing the VDFs at different times (see Supplementary Video

C) shows that both ions usually resemble gyrating ion populations.

In light of the complex simulated VDF shapes, and in order to get a better understanding of global foreshock ion characteristics, we show in Figure 8 plots of global per-species temperature anisotropies and a measure of per-species non-gyrotropy. We apply the temperature anisotropy to the whole VDF allowing us to identify regions where VDFs have FAB-like features, showing up as anisotropy values much smaller than 1. The agyrotropy measure (Swisdak, 2016)

$$Q_{\mathrm{ag}} = \frac{P_{12}^2 + P_{13}^2 + P_{23}^2}{P_\perp^2 + 2P_\perp P_\parallel}, \tag{1}$$

where $P_\parallel = P_{11}$ is the parallel pressure, $P_\perp = 0.5(P_{22} + P_{33)}$ is the perpendicular pressure, and $P_{12}$, $P_{13}$, and $P_{23}$ are off-diagonal pressure tensor components, can be used to evaluate the complexity of the VDF and the role of gyrating ions. For a completely gyrotropic distribution $Q_{\mathrm{ag}} = 0$, and $Q_{\mathrm{ag}} = 1$ would signify a maximal deviation from gyrotropy. Panels (a) and (b) of Figure 8 show agyrotropies for protons and helium, respectively, and panels (c) and (d) temperature anisotropies. The





panels in Figure 8 also show out-of-plane magnetic field contours at a level of $\pm 0.2\,\mathrm{nT}$, indicating the extent and structure of the ULF foreshock. We find that the FAB region of protons is clearly visible in panel (c) at the edge of the ion foreshock as a dark orange band ($T_{\perp,\mathrm{p}}T_{\parallel,\mathrm{p}}^{-1} \approx 0.2$). In panel a) we see the same structure at the foreshock edge. It is visible as a band of medium green blobs ($Q_{\mathrm{ag,p}} \sim 0.01$) close to the shock nose, transitioning to paler green thin streaks ($Q_{\mathrm{ag,p}} \sim 0.001$) away from the shock.

For helium, in panel d), this FAB region at the edge of the foreshock is also clearly visible, with very low anisotropy values. Interestingly, helium also shows a parallel pressure signature (dark orange low anisotropy values) deeper in the foreshock. This region coincides with a weakening of the ULF foreshock, visible in the de-structuring of wave fronts (black contours) in the vicinity of ($X = 10\,R_{\mathrm{E}}, Z = -40\,R_{\mathrm{E}}$). Referring back to Figure 1b we see that this band of low-energy FAB-type alpha-particles coincides with an increase of measured helium fraction, which continues downstream from that point.

Panels a) and b) of Figure 8 show, in particular for protons, an enhancement in agyrotropy at the boundary between the FAB-region and the ULF foreshock. This dark-green region is a signature of gyrating and gyrophase-bunched ions at this boundary, in agreement with Meziane et al. (2004), Mazelle et al. (2007), and Andrés et al. (2015). This effect can also be seen at virtual spacecraft location B in Figure 6, panel (2), subpanels [p⊥⊥] and [$\alpha$ ⊥⊥], as a gyrophase-bunched extension of the ion distribution. We note that this increase of agyrotropy and thus gyrating ions matches the ULF foreshock and previous

studies well down to about $Z = -25\,R_{\mathrm{E}}$, but the boundary becomes less well defined further out, away from the shock. We also note that panel b) shows darkened bands of increased agyrotropy on both upstream and downstream edges of the inner foreshock heightened parallel pressure alpha-particle band, which suggests there might be similar gyrophase sampling taking place as for the foreshock FAB beam proper.

    For both protons and helium, we see striped enhancements of agyrotropy right at the outer ion foreshock boundary, indicative

of spatial sampling of gyrating ion beams right at the outermost foreshock edge. The gyration of these ion beams, accelerated at the quasi-perpendicular shock front and made non-uniform by the rippling of the shock front, are particularly visible in Supplementary Video D, which is an animated version of Figure 8. Finally, we note that large regions of the foreshock close to the quasi-parallel bow shock show signatures of temperature anisotropies $\gtrsim 1$ and enhanced agyrotropies, which are likely a signature of specular reflection of ions at the quasi-parallel bow shock.

### 3.3   Foreshock waves

    Figures 9 and 10 show measurements from virtual spacecraft placed at locations C, D, E and F in the foreshock (see Figure 6 and 7). The top row in each plot displays the total number density $n_{\mathrm{tot}}$ for protons and helium, and the second row displays their suprathermal number densities $n_{\mathrm{st}}$. On both rows, the helium number densities have been multiplied by a factor of 100 to ease the comparison with proton number densities. Note that the suprathermal number densities are shown on a logarithmic

scale, since the values vary significantly. In the third row, temperature anisotropy $T_{\perp}T_{\parallel}^{-1}$ is displayed, and the fourth row displays the agyrotropy measure $Q_{\mathrm{ag}}$. The time series of $Q_{\mathrm{ag}}$ have been smoothed out with a 5 second running average to help readability due to a large amount of high-frequency fluctuations. The fifth and sixth rows display the total magnetic field $|B|$ and its out-of-plane component $B_y$, respectively.





The fluctuations of $H^+$ and $He^{2+}$ total densities follow each other quite closely at all locations, though the amplitude of

315 the $He^{2+}$ density oscillations is larger than that of protons at point D and E. These larger $He^{2+}$ density variations seem to be well correlated with the fluctuations of the $He^{2+}$ suprathermal density at position D (Figure 9j), but not so much at point E (Figure 10b). At all locations analyzed with these virtual spacecrafts, the suprathermal ion ratio $n_{\alpha,\mathrm{st}} n_{\mathrm{p,st}}^{-1}$ is larger than the solar wind ion ratio $n_{\alpha,\mathrm{sw}} n_{\mathrm{p,sw}}^{-1}$, as shown already by Figure 1b. The agyrotropy is also more pronounced for $He^{2+}$ than for $H^+$, as illustrated in the top panels of Figure 8 and with the VDF discussed in the previous section.

The two bottom rows of Figures 9 and 10 display the wavelet power spectra of $|B|$ and $B_y$, in order to investigate foreshock wave activity. The wavelet transform (Torrence and Compo, 1998) is calculated using the Morlet wavelet. The black contours in the power spectra show the 95% confidence level and the cross-hatched regions bordering the spectra depict the "cone of influence", where edge effects arising due to the time series' endpoints become important. The horizontal dashed line on the $B_y$ wavelet power spectra is drawn at 50 s, which is close to the expected spacecraft frame period of foreshock fast magnetosonic

waves for these upstream conditions according to empirical models (Le and Russell, 1996; Takahashi et al., 1984). At all four positions, an enhancement in the $B_y$ component power spectra can be seen in the vicinity of this period, in agreement with previous works showing that fast magnetosonic waves permeate the foreshock in Vlasiator simulations as in spacecraft observations (Palmroth et al., 2015; Turc et al., 2018). We note that none of the virtual spacecraft selected here display the typical quasi-monochromatic foreshock waves shown in these previous studies, due to their relative proximity to the bow

shock ($\lesssim 7\,R_{\mathrm{E}}$). In the vicinity of the shock, the wave activity is more complex due to nuanced interactions between the waves and the diffuse ion population (Greenstadt et al., 1995; Turc et al., 2018).

At point F, we find weaker wave activity, despite its location deep within the foreshock. This is most likely due to the low density of suprathermal gyrating or beam particles in this part of the foreshock (see Figure 10j). This results in a lower wave growth rate, as this parameter depends on the beam density for the beam-beam instabilities at play in the foreshock (Gary,

1993). This weaker wave activity is accompanied by a lower temperature anisotropy for the $He^{2+}$ ions, due to the suprathermal $He^{2+}$ population being in the form of low-energy field-aligned beams in this region (see panel (4) of Figure 7). Comparing the positions of our virtual spacecraft with Figure 8 and the contours indicating well-structured or more broken up ULF wave fronts, we see that the weakest wave power (point F) is seen at the most broken-up position, and the strongest power (point D) is found at a position of well-structured wave fronts.

According to previous works, the plasma rest frame frequency of foreshock fast magnetosonic waves generated by proton beams is of the order of 10% of the proton gyrofrequency, probably due to the cyclotron resonance which gives rise to the waves (Hoppe and Russell, 1982; Eastwood et al., 2005; Wilson III, 2016). $He^{2+}$ ions have a gyrofrequency that is half that of protons. $He^{2+}$ ion beams could thus generate fast magnetosonic waves at a period of $\sim 100\,\mathrm{s}$ with the upstream parameters used in our simulation. The wavelet power spectra in Figures 9 and 10 show enhanced wave power around $\sim 100\,\mathrm{s}$, but this

part of the spectra is mostly inside the cone of influence of the wavelet transform, making it difficult to draw firm conclusions. Moreover, the analysis of the foreshock wave properties in an identical Vlasiator run without a helium population (not shown) reveals similar enhancements of the wave power at $\sim 100\,\mathrm{s}$. It is therefore unlikely that these fluctuations are due to helium





beam instabilities. Thus, despite a 1% solar wind helium content providing non-negligible mass loading, we find the helium component to not have a significant impact on foreshock wave populations.

## 4   Discussion and conclusions

In this study, we present how sometimes the proton foreshock extends further out upstream than the helium foreshock. Our model of the edge of the foreshock shows that within a few $R_\mathrm{E}$ of the foreshock edge, the ratio of suprathermal alphas to protons (as shown in Figure 1b) often tends to a value of about 10 times less than the solar wind. This seems to be in excellent agreement with Figure 1b of Fuselier and Thomsen (1992). They restricted the analysis of beam ions to high energies ($1.6-5.7$ times the solar wind energy), which will likely require measurements very close to the foreshock edge. At any given distance from the shock, slower particles will have travelled for a greater period of time and will thus have drifted deeper into the foreshock due to, e.g., $\mathbf{E} \times \mathbf{B}$ drift. In the central section of the FAB region but still outside the ULF foreshock we report a suprathermal alpha fraction of the order of the solar wind value. At the very sunward edge of the foreshock in Figure 1b some regions of relatively more abundant suprathermal alphas can be seen, and we suggest that this is an effect stemming from the larger gyroradii of alpha-particles, as the feature is clearly spatially periodic.

In our simulation, we find that well inside of the upstream edge of the foreshock there is a region with strong, well-structured ULF wavefronts and a relative decrease of suprathermal alpha-particles, as shown in Figure 1b. We suggest that this could be due to these very strong proton-driven ULF waves being strong enough to scatter even alpha-particles, which are not in resonance with the wave oscillation due to a larger mass-to-charge ratio than that of protons. Further inside the foreshock the ULF wave front breaks up into less uniform waves. This breakup is inherent to proton dynamics and not caused by the presence of alphas. Previously, growth of waves with distance from the foreshock edge has been reported in Le and Russell (1992), but their study was performed close to the nose of the shock, whereas the de-structuring we report takes place far along the flank of the bow shock. In this region we report a helium fraction higher than that of the solar wind. One possible explanation for this is that the destructured ULF waves are still able to scatter protons, but alpha-particles propagate in a less disturbed fashion, more akin to a low-energy FAB. This is seen as a second region of parallel pressure enhancement for alphas in Figure 8d.

Based on the time-energy spectrogram results in panels c) and d) of Figures 3 and 4, it appears that the foreshock edge transitions for $H^+$ and $He^{2+}$ were in more agreement for the December 2018 than for the November 2018 event. During the event depicted in Figure 3, the IMF upstream of the foreshock was dominated by the $B_y$-component, and MMS was located at approximately $X = 13, Y = 1, Z = 5 R_\mathrm{E}$ (roughly at the nose of the shock). If we assume a parabolic bow shock shape, the field lines at the foreshock edge were thus connected to a region of the shock where $\theta_{B\mathrm{n}} \sim 45°$. For each field line, we can estimate the derivative of $\theta_{B\mathrm{n}}$

$$\frac{d\theta_{B\mathrm{n}}}{d\tilde{r}_{\perp\mathrm{FSedge}}}$$

where $\tilde{r}_{\perp\mathrm{FSedge}}$ is a normalized spatial distance vector perpendicular to the foreshock edge. At the location of MMS with the listed IMF conditions, the derivative of $\theta_{B\mathrm{n}}$ respective to the distance to the foreshock edge is large. Conversely, during the





event depicted in Figure 4, MMS was located at approximately $X = 10, Y = 11, Z = 6R_{\mathrm{E}}$, i.e. somewhat at the flank, and the
IMF had a strong $-B_x$ component. Thus, the field lines at the foreshock edge can be assumed to connect to a region where
the corresponding derivative of $\theta_{B\mathrm{n}}$ for each field line is small. When comparing two Vlasiator simulations with different
IMF directions in Figure 5 and corresponding qualitatively with the MMS observation situations, we see qualitatively similar
behaviour. We note that the contours in Figure 5 were averaged over time, smoothing out variations due to ion gyroradii, ion

gyrotimes, and reflection variations due to mesoscale bow shock reformation. Variations such as these are detectable in virtual
and real spacecraft measurements. Based on this comparison, we suggest that the different profiles of the foreshock edge
transition, seen particularly well in the suprathermal ion densities, may be related to the derivative of $\theta_{B\mathrm{n}}$ for the connecting
field line, as we are able to replicate the MMS observation variation in the gradient of suprathermal ion profiles using two
simulation runs with different IMF cone angles. In a previous study, Sibeck et al. (2008) investigated foreshock edge gradients

in radial IMF hybrid simulations, finding a correlation with model dipole tilt. They stated that their bow shock edges were
still propagating outwards, and the foreshock in their Figure 1 appears tilted to the south, possibly as a result of non-radial
magnetic field components and perhaps a similar underlying cause as in our explanation. This can be investigated further in
both observational and simulational studies.

Figure 8 shows that helium exhibits a greater agyrotropy throughout the ion foreshock. Interestingly, enhanced helium

agyrotropy appears to coincide with well-structured ULF regions. This suggests that the mostly proton-induced ULF waves
might be efficient drivers of agyrotropy for gyrating $He^{2+}$ ions which have half the gyrofrequency of protons, leading them to
scatter into the diffuse distribution. Proton-induced waves have been shown to heat the solar wind helium populations (Hollweg
and Turner, 1978; Dusenbery and Hollweg, 1981). We propose that this additional heating by ULF waves is one reason for
helium ions exhibiting greater agyrotropy values than protons throughout the foreshock. Another source of agyrotropy and

VDF break-up in helium may be that $He^{2+}$ ions have greater gyroradii than $H^+$ ions, and thus each ion scans a greater extent
of foreshock waves and structures.

We note the absence of a diffuse population of ions in our proton and helium VDFs in Figures 6 and 7, but deduce that
it is due to the low phase-space density of the diffuse ion population extending below our velocity space sparsity threshold,
leading to those ions being discarded. We do not expect the diffuse ion population to strongly affect the wave dynamics,

maintaining reasonable validity of the rest of our results. We do admit that some of our deductions about proton and alpha-
particle scatterings depend on this assumption that the suprathermal density decrease can be interpreted as scattering of ions
from a gyrating population to a diffuse population. At least some of the reported relative alpha-particle abundance in Figure 1
can be attributed to this, which agrees with the proton-induced ULF waves being more efficient at scattering protons, leading
to a greater portion of the proton suprathermal population scattering into the diffuse part of velocity space, whereas alphas

remain as gyrating ions tracked by the simulation (and resulting in high agyrotropies as well). We also note that our study is
limited to lower-energy FABs with energies of the order of $1 - 1.5$ times the solar wind energy. The lack of a deka-keV FAB
in our simulation may result from our choice of cold electrons i.e. no electron pressure gradient term at the shock front. Since
protons with a mass-to-charge ratio of 1 would feel the effect of a cross-shock potential stronger than alpha-particles would,
this can also partially explain the predominantly large helium fraction in our simulated foreshock.





We evaluate virtual spacecraft wavelets, looking for signatures of waves driven by helium beam instabilities which could develop together with the proton beam instabilities in the foreshock, but do not find convincing evidence for such waves. This result is as expected because of the low abundance of $He^{2+}$ ions in our simulation, only 1% of the solar wind proton number density. This number density is still enough to cause mass loading, pushing the bow shock $0.5 - 1.0\,R_E$ further towards the Earth than in a comparative run without helium (comparison not shown here, for the other simulation see Blanco-Cano et al. 2018). Future simulations with a higher helium to proton ratio will allow further investigation of helium-driven waves in the foreshock. We note that due to the ratio of gyrofrequencies, simulations must be run for an extended period of time in order to accurately capture potential helium-induced waves.

Providing a point of comparison with a similar model, Jarvinen et al. (2019) simulated the magnetosphere and the foreshock of Mercury with a global hybrid cloud-in-cell model, including $4\%$ of $He^{2+}$ ions in solar wind. Jarvinen et al. (2019) presented a fast-mode ULF wavefield at wave periods of approximately $5$ s, and populations of both $H^+$ and $He^{2+}$ were present in the quasi-parallel bow shock region, with $He^{2+}$ backscattering found to be more efficient than for $H^+$. Similar to our model, they did not include an electron pressure gradient term. We note that despite how the Hermean magnetosphere is much smaller and the impinging solar wind is quite different from that at the Earth, some properties of ULF waves at different planetary foreshocks appear to scale with respect to the interplanetary magnetic field magnitude, as illustrated by Hoppe and Russell (1982). Additionally, panel b) of our Figure 1 shows proportionally enhanced suprathermal $He^{2+}$ densities compared to $H^+$, which is similar to the point result in Jarvinen et al. (2019), but we also show spatial structure in the ratio of the suprathermal populations.

In summary, we show how for the simulated solar wind conditions, $He^{2+}$ is a significant foreshock species which has dynamics similar to those of protons but with distinct properties such as preferential heating by proton-induced ULF waves. Both protons and helium are found in the FAB at the foreshock edge, with beam energies decreasing from $\gtrsim 5\,keV/nuc$ to $\sim 3\,keV/nuc$ when going from the foreshock edge to the inner edge of the FAB region. Helium ions are found to exhibit more agyrotropy than protons, probably due to their interaction with the proton-dominated ULF waves.

The profiles of suprathermal abundances of proton and helium at the foreshock edge show variability both in Vlasiator simulations and in MMS data, and we show how this may be explained with the prevailing IMF orientation affecting the derivative of $\theta_{Bn}$ at the field-line-connected position at the bow shock.

*Code and data availability.* Vlasiator (http://www.physics.helsinki.fi/vlasiator/, Palmroth, 2020) is distributed under the GPL-2 open source license at https://github.com/fmihpc/vlasiator/ (Palmroth and the Vlasiator team, 2020). Vlasiator uses a data structure developed in-house (https://github.com/fmihpc/vlsv/, Sandroos, 2019). The Analysator software (https://github.com/fmihpc/analysator/, Hannuksela and the Vlasiator team, 2020) was used to produce the presented figures. The run described here takes several terabytes of disk space and is kept in storage maintained within the CSC – IT Center for Science. Data presented in this paper can be accessed by following the data policy on the Vlasiator web site.



*Video supplement.* The Supplementary Videos A, B, C, and D provide movie extensions of Figures 1b, 6, 7, and 8, showcasing the temporal evolution of foreshock features and particle population shapes and properties.

Movie A (Battarbee et al., 2020a) is a movie extension of panel b of Figure 1. Animation of the foreshock region of the simulation over 250 seconds of simulation. Ratio of suprathermal densities for helium over proton, normalized to the solar wind ratio of 1%. The ratio is not shown in the pristine solar wind. Black contours are drawn for $B_y = \pm 0.1\,\mathrm{nT}$.

Movie B (Battarbee et al., 2020b) is a movie extension of Figure 6. Animation of velocity distribution functions for protons and alpha-particles and their locations in the foreshock. Panel (3): Map of the foreshock with three VDF positions indicated with capital letters. The colour indicates proton temperature anisotropy with magnetic field lines in black. Panels (1), (2), and (4): Sets of four projections of ion
VDFs at virtual spacecraft locations in the solar wind frame. Coordinates are chosen to represent two positions at the foreshock boundary (A, B) and one just within the ULF foreshock (C). In each set, subpanels are labelled as $v_B$ vs $v_{B\times(B\times V)}$ ($\|\perp$, left columns) or $v_{B\times V}$ vs $v_{B\times(B\times V)}$ ($\perp\perp$, right columns) and protons (top rows) or helium (bottom rows).

Movie C (Battarbee et al., 2020c) is a movie extension of Figure 7. Animation of velocity distribution functions for protons and alpha-particles and their locations in the foreshock. Panel (3): Map of the foreshock with three VDF positions indicated with capital letters. The
460 colour indicates proton temperature anisotropy with magnetic field lines in black. Panels (1), (2), and (4): Sets of four projections of ion VDFs at virtual spacecraft locations in the solar wind frame. Coordinates are chosen to represent two positions close to the quasi-parallel bow shock (D, E) and one deep within the outer foreshock (F). In each set, subpanels are labelled as $v_B$ vs $v_{B\times(B\times V)}$ ($\|\perp$, left columns) or $v_{B\times V}$ vs $v_{B\times(B\times V)}$ ($\perp\perp$, right columns) and protons (top rows) or helium (bottom rows).

Movie D (Battarbee et al., 2020d) is a movie extension of Figure 8. Animation of foreshock characteristics for proton and helium over
465 363 seconds of simulation. The left column (panels a and c) show protons, the right column (panels b and d) helium. Top row: agyrotropy $Q_{\mathrm{ag}}$ (see eq. 1) where a value of 0 corresponds with perfect gyrotropy. Bottom row: Temperature anisotropy $T_\perp T_\|^{-1}$, calculated for the total distribution function including both solar wind and suprathermal parts. Black contours show the out-of-plane magnetic field $B_y$ fluctuations at a level of $\pm 0.2\,\mathrm{nT}$, indicating the extent of the ULF foreshock region.

*Author contributions.* This paper was outlined and drafted in the Third International Vlasiator Science Hackathon held in Helsinki, 19-23
August 2019. All co-authors provided scientific input either during the Hackathon event, during the pre-submission review process, or both. MB, XBC, LT, and PK performed preliminary investigation leading to the team effort. MB led the investigation and the writing process. AJ provided MMS plots and SF and KT verified them. VT performed wavelet analysis with the help of LT. MP leads the Vlasiator team and organized the hackathon. UG and YPK were instrumental in adding helium support to Vlasiator.

*Competing interests.* The authors declare that they have no conflict of interest.

*Acknowledgements.* This paper was outlined and drafted in the Third International Vlasiator Science Hackathon held in Helsinki, 19-23 Aug 2019. The Hackathon was funded by the European Research Council grant 682068 – PRESTISSIMO. We acknowledge the European Research Council for Starting grant 200141-QuESpace, with which Vlasiator was developed, and Consolidator grant 682068-PRESTISSIMO



awarded to further develop Vlasiator and use it for scientific investigations. The Finnish Centre of Excellence in Research of Sustainable Space, funded through the Academy of Finland grant number 312351, supports Vlasiator development and science as well. We also grate-
fully acknowledge the Academy of Finland (grant numbers 322544, 328893 and 309937). XBC acknowledges the UNAM PAPIIT-DGAPA project (IN105218-3) and CONACyT (255203) grants. MAT acknowledges the NASA MMS contract NNG04EB99C, NASA MMS GI grant 80NSSC18K1363, NASA grant 80NSSC18K0999, and the DGA project, École Polytechnique, convention 2778/IMES. PK's work is supported by the PAPIIT grant IA101118. Research at Southwest Research Institute was supported by the MMS prime contract NNG04EB99C.

This research was made possible with the data and efforts of the people of the Magnetospheric Multiscale mission. The data are available
through the MMS Science Data center[2], and the Coordinated Data Analysis Web (CDAWeb)[3].

---

[2]https://lasp.colorado.edu/mms/sdc/public/
[3]https://cdaweb.sci.gsfc.nasa.gov/index.html/



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



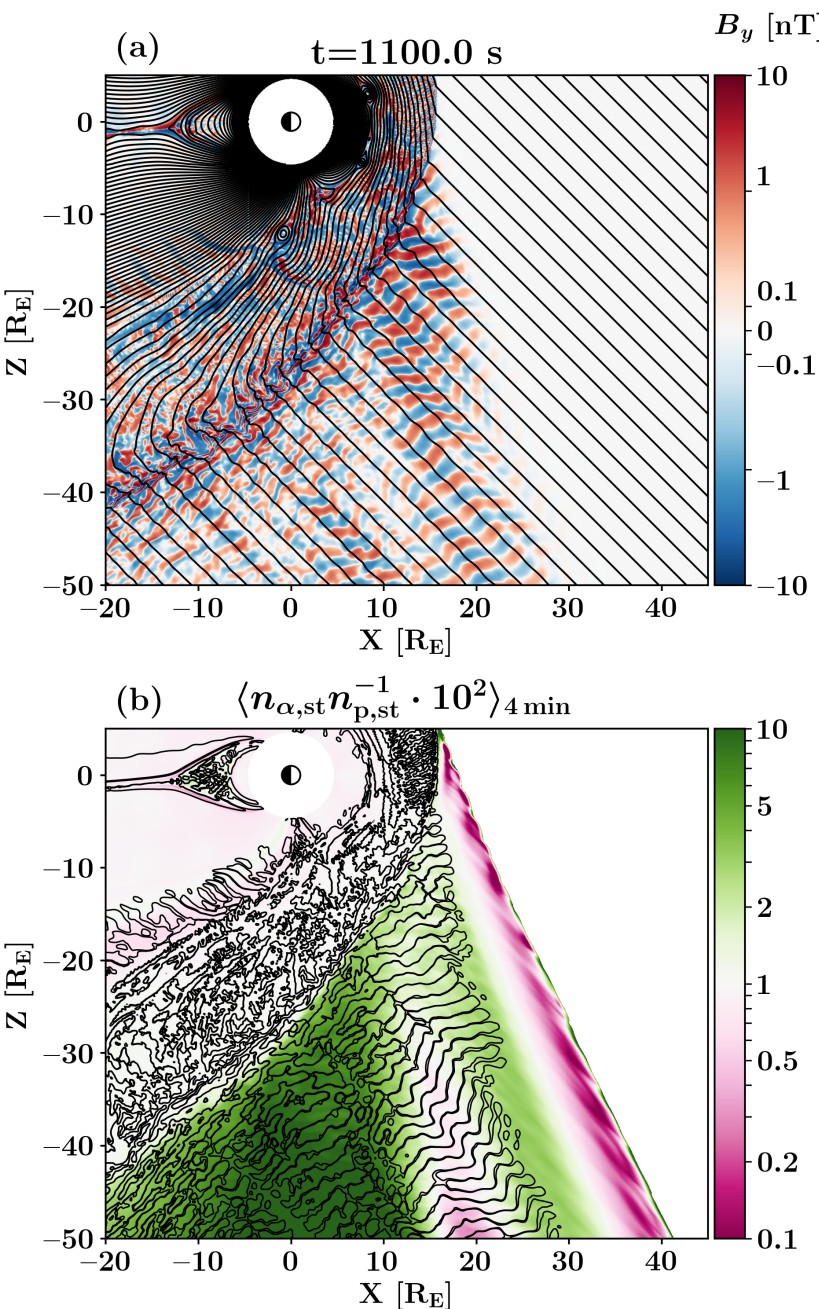

**Figure 1.** Overview of the foreshock region of the simulation. Panel a: Out-of-plane magnetic field fluctuations $B_y$ at 1100 seconds using a symmetric logarithmic colour scale, indicating the extent of the ULF foreshock region. Black curves indicate magnetic field lines. Panel b: Ratio of suprathermal densities for helium over proton, normalized to the solar wind ratio of 1%, averaged over a period of 4 minutes. The ratio is not shown in the pristine solar wind. Black contours are drawn for $B_y = \pm 0.1\,\mathrm{nT}$ at time $t = 1100\,\mathrm{s}$.





**Figure 2.** Left panel a: Magnetic field out-of-plane component $B_y$ indicating the ULF foreshock extent. Contours indicate proton (black) and helium (green) suprathermal densities at 0.1 % (dotted contours) and 0.5 % (solid contours) of solar wind values. Four thick lines indicate cut-throughs across the foreshock boundary. Right panels b-e: profiles across the foreshock at the positions shown in panel a. Profiles are shown for suprathermal proton density (black solid), suprathermal helium density (green solid), and magnetic field magnitude (blue solid). The horizontal dashed lines indicate 0.5 % (dashed) of the solar wind density.

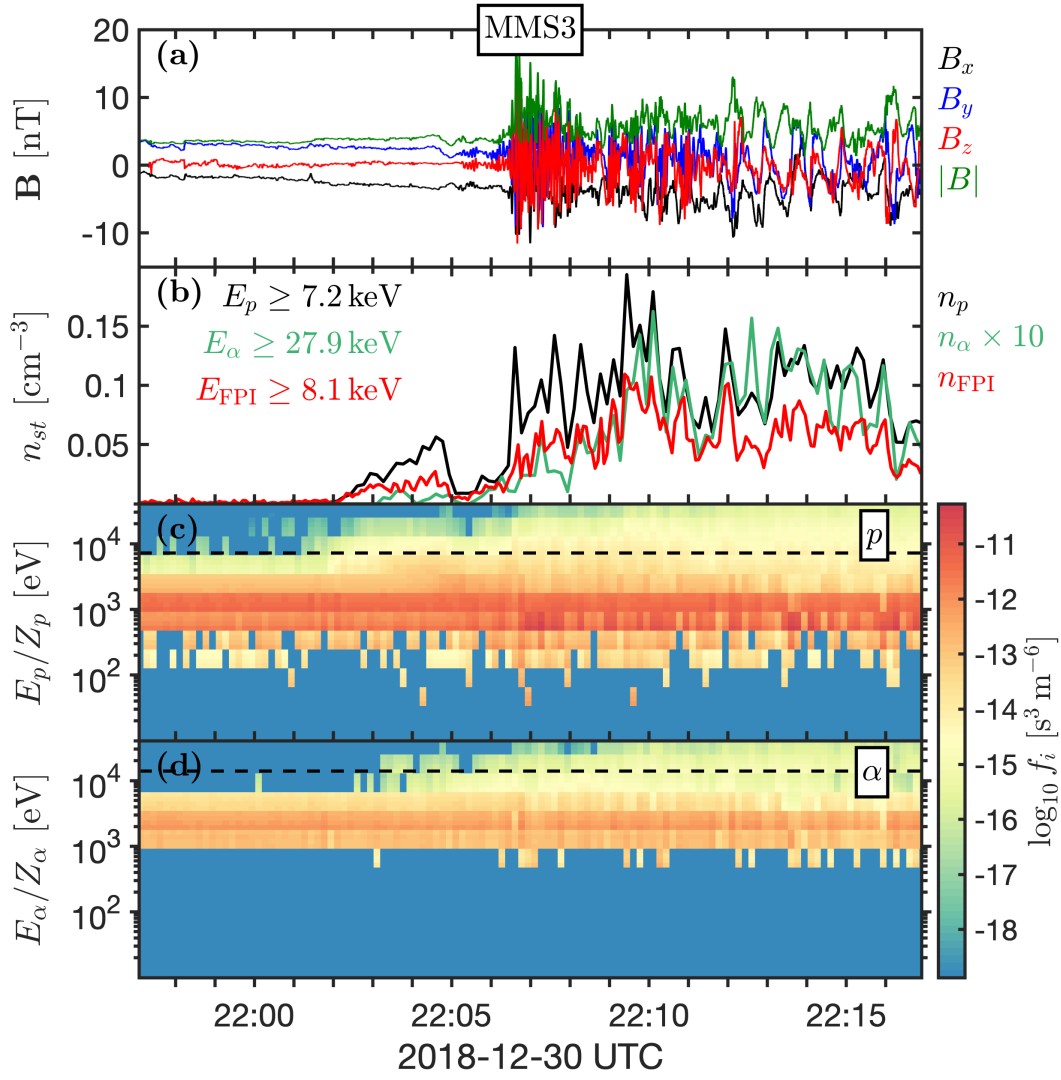

**Figure 3.** MMS3 crossing from the foreshock region into the solar wind on 30 December 2018. Panel a) shows survey mode measurements of magnetic field magnitude and components, showing the spacecraft exiting the ULF foreshock. Panel b) displays number densities for H⁺ and He²⁺ suprathermal ions measured by HPCA with spacecraft frame energy cutoffs of 2.5 keV per nucleon, with the helium density multiplied by 10 to ease density evaluation. FPI ion number density is shown for comparison. Panels c) and d) show HPCA ion energy spectrograms for protons and helium with the dashed line indicating the respective suprathermal ion cutoff energies.

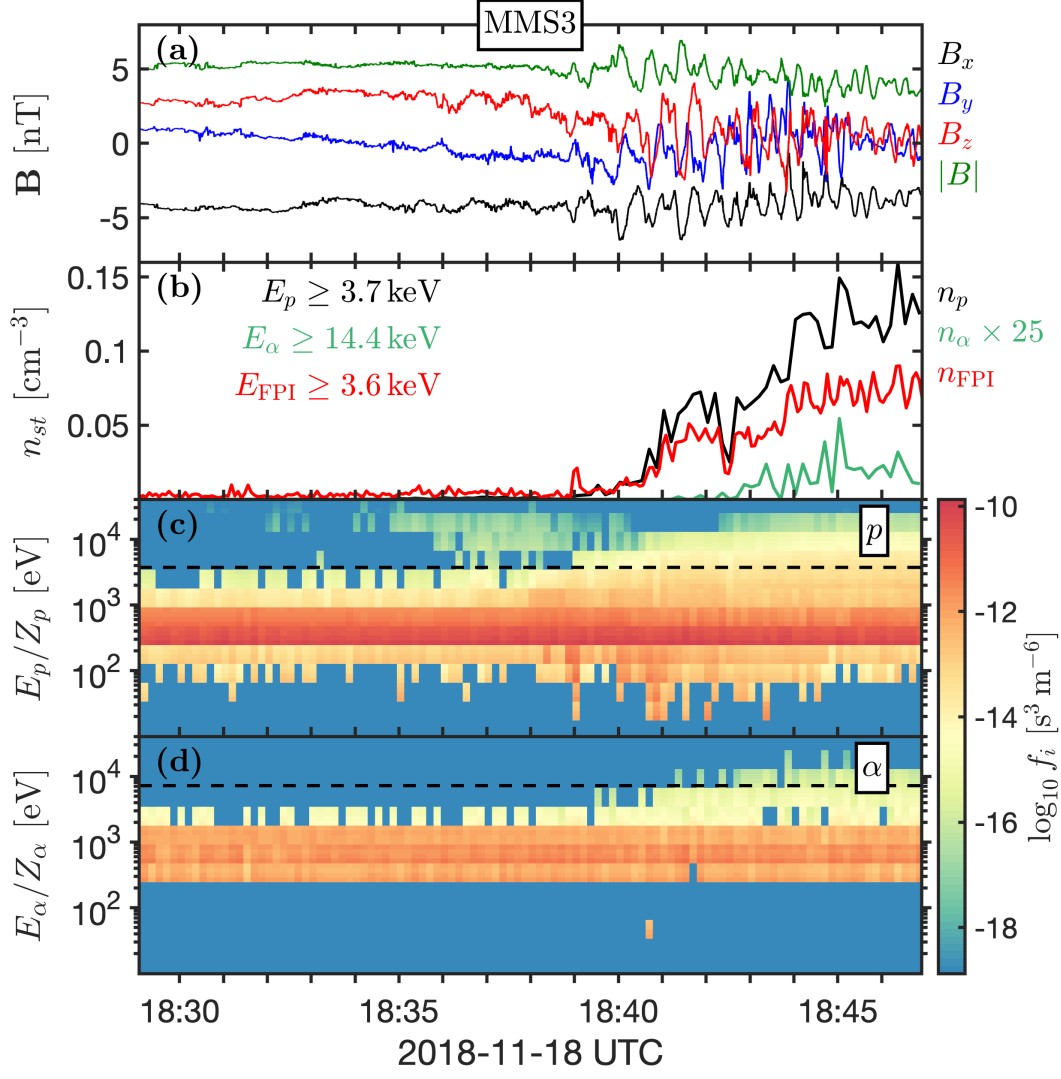

**Figure 4.** MMS3 crossing from the solar wind into the foreshock on 18 November 2018. The format is the same as in the Figure 3 but with
$He^{2+}$ multiplied by 25.





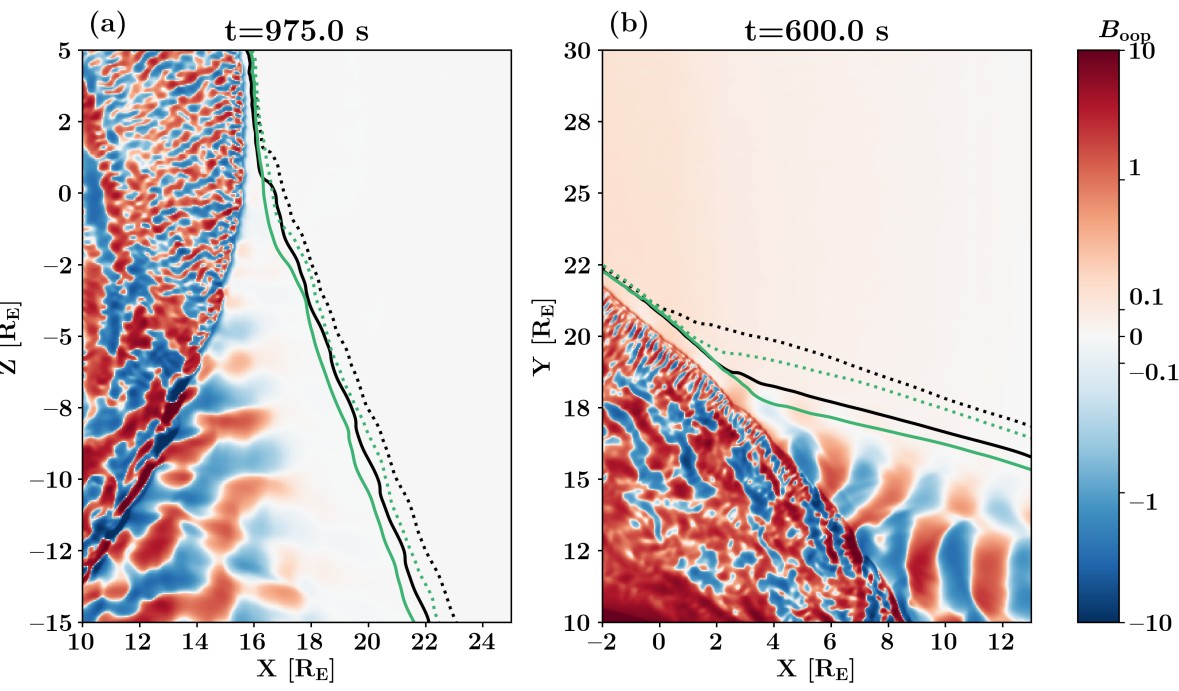

**Figure 5.** A zoom-in of the foreshock edge with magnetic field out-of-plane component $B_{\mathrm{oop}}$ indicating the ULF foreshock extent. Contours indicate suprathermal ion densities as in Figure 2a but averaged over 2 minutes. Left panel (a): the meridional plane $45°$ IMF simulation used in the majority of this paper. Right panel (b): An equatorial plane $5°$ IMF run for comparison. IMF orientation appears to affect the suprathermal ion density gradient at the foreshock edge.



**Figure 6.** Velocity distribution functions for protons and alpha-particles and their locations in the foreshock. Panel (3): Map of the foreshock with three VDF positions indicated with capital letters. The colour indicates proton temperature anisotropy with magnetic field lines in black. Panels (1), (2), and (4): Sets of four projections of ion VDFs at virtual spacecraft locations in the solar wind frame. Coordinates are chosen to represent two positions at the foreshock boundary (A, B) and one just within the ULF foreshock (C). In each set, subpanels are labelled as $v_B$ vs $v_{B\times(B\times V)}$ ($\|\perp$, left columns) or $v_{B\times V}$ vs $v_{B\times(B\times V)}$ ($\perp\perp$, right columns) and protons (top rows) or helium (bottom rows). Black-and-white circles estimate areas of specular reflection.







**Figure 7.** Velocity distribution functions for protons and alpha-particles and their locations in the foreshock. Panel (3): Map of the foreshock with three VDF positions indicated with capital letters. The colour indicates proton temperature anisotropy with magnetic field lines in black. Panels (1), (2), and (4): Sets of four projections of ion VDFs at virtual spacecraft locations in the solar wind frame. Coordinates are chosen to represent two positions close to the quasi-parallel bow shock (D, E) and one deep within the outer foreshock (F). In each set, subpanels are labelled as $v_B$ vs $v_{B\times(B\times V)}$ ($\|\perp$, left columns) or $v_{B\times V}$ vs $v_{B\times(B\times V)}$ ($\perp\perp$, right columns) and protons (top rows) or helium (bottom rows). Black-and-white circles estimate areas of specular reflection.



**Figure 8.** Foreshock characteristics for proton and helium at 1100 seconds. The left column (panels a and c) show protons, the right column (panels b and d) helium. Top row: agyrotropy $Q_{\mathrm{ag}}$ (see eq. 1) where a value of 0 corresponds with perfect gyrotropy. Bottom row: Temperature anisotropy $T_\perp T_\parallel^{-1}$, calculated for the total distribution function including both solar wind and suprathermal parts. Black contours show the out-of-plane magnetic field $B_y$ fluctuations at a level of $\pm 0.2\,\mathrm{nT}$, indicating the extent of the ULF foreshock region.







**Figure 9.** Time series and wavelets at virtual spacecraft positions C (panels a to h) and D (panels i to p) at the edge of the ULF foreshock. Panels a)-d) and i)-l) show proton (helium) quantities in black (green). a) and i): total number densities. b) and j): suprathermal number densities on a logarithmic scale. The helium number densities have been multiplied by 100 in panels a), b), i), j). Panels c) and k): temperature anisotropies. d) and l): 5 second running averages of agyrotropy $Q_{\mathrm{ag}}$. e) and m): Magnetic field magnitudes $|B|$. f) and n): Magnetic field out-of-plane components $B_y$. g) and o): Wavelet power spectra of $|B|$ fluctuations. h) and p): Wavelet power spectra of $B_y$ fluctuations. The black contours show the 95% confidence level and the cross-hatched regions denote the "cone of influence". The white dashed lines show the expected frequency of $B_y$ fluctuations (see text).





**Figure 10.** Time series and wavelets at virtual spacecraft positions E (panels a to h) and F (panels i to p) in the ULF foreshock. Panels a)-d) and i)-l) show proton (helium) quantities in black (green). a) and i): total number densities. b) and j): suprathermal number densities on a logarithmic scale. The helium number densities have been multiplied by 100 in panels a), b), i), j). Panels c) and k): temperature anisotropies. d) and l): 5 second running averages of agyrotropy $Q_{ag}$. e) and m): Magnetic field magnitudes $|B|$. f) and n): Magnetic field out-of-plane components $B_y$. g) and o): Wavelet power spectra of $|B|$ fluctuations. h) and p): Wavelet power spectra of $B_y$ fluctuations. The black contours show the 95% confidence level and the cross-hatched regions denote the "cone of influence". The white dashed lines show the expected frequency of $B_y$ fluctuations (see text).