# Peer review of "Helium in the Earth's foreshock: a global Vlasiorator survey"

_Annales Geophysicae, 2020_

## Referee Comment (RC1) · Jeffrey Broll (Referee) · 27 Jun 2020

The submitted work investigates the helium foreshock in simulation and MMS/HPCA observation. (Throughout, all mentioned helium is doubly ionized.)

The main simulation of interest is 2D3V hybrid-Vlasov ions and massless-fluid electrons, with setup very similar to previously published work - low-$\beta$ fast solar wind, 45° cone-angle, supercritical. This is as appropriate for the study as is feasible, but I must wonder what acts in place of electron pressure gradients for the cross-shock potential. I suspect/guess (cf. I think Gosling+ 82, "Evidence for spec. refl. ions upstr. from the Q$\parallel$ bow shock", and Gosling + 89, "Ion refl. and downstream thermalization at the Q$\parallel$ bow shock", it's been a while) that some deviation from the specular reflection ovals should

be expected if the main shock ramp jump is smaller or less sharp, in particular - this is possibly a shortcoming of the reviewer, but some demonstration of the cross-shock potential at an interesting point or two would have been nice to see...

The data used are appropriate for the study (although more could have been gotten out of burst HPCA data as in the cited HPCA papers). However it's not clear if the figures couldn't be better labeled - Figures 3's caption is traveling backwards in time. Further, the suprathermal proton and helium population seem to go above HPCA's energy range on 30 Dec 2018 so perhaps some kind of either Maxwellian or $\kappa$ fit to averaged foreshock E/Z slice would increase confidence that the suprathermals were being counted far enough up. This issue and the different suprathermal definitions (which could be explained in the text for those not fluent in Vlasiator nuances) seem to make comparison harder than necessary.

The analysis lost me a bit right around the very large $T_\parallel/T_\perp$ — could the resolution be hindering something like firehose growth that would bring that down? This could be an issue; checking Gary's microinstabilities book (7.2.1 and fig. 7.1) and assuming $\beta$ sane, I'd guess that you could be missing some physics there. I think that given the nature of the work this is to be expected, but it could be mentioned for completion's sake.

Having picked those nits I think this work should be published after some minor-to-moderate complaints are addressed. In general there are some Vlasiator-specific nuances that are not common enough knowledge (at least for this reviewer) and possibly could be made so without detracting from the simulations section.

J Broll,

Los Alamos

---

## Referee Comment (RC2) · David Sibeck (Referee) · 29 Jun 2020

This paper presents a comprehensive analysis of novel results from the Vlasiator model for proton and helium acceleration in the Earth's foreshock. The paper is clear and (in general) well written, the conclusions are substantiated by discussion of simulation results and comparison with observations. For the community interested in shock physics, the paper will be very important. For the general space plasma physicist., the results will be fairly important. Some of the results reported in the paper include (1) different edge locations for the proton and helium foreshock, (2) the manner in which the ratio of helium to proton density varies with location in the foreshock, (3) the nature of proton and helium distribution functions as a function of location, (4) the manner by which helium is heated in the foreshock, (5) the effects of the IMF orientation on fore-

shock boundary structure, and (6) the nature of waves/turbulence in and around the foreshock.

I only have a few comments/questions.

Lines 50-61. I did not find this review of past work as clear as it could be. I have no objection to each sentence but I think it can be presented more carefully. The authors could systematically go through each region of space, or each type of distribution function, showing they are covering all possibilities. A table noting regions, types of distribution functions, and composition ratios would help. Could the authors just tell what is seen first, and then give explanations? Or could they state expectations and then tell what past work has seen? It would be cleaner than the mixture of observations and interpretations.

Having a table would also be something useful that the authors could refer back to when summarizing their work towards the end of the paper, especially if they can check off each observation and state that their model predicted it.

Change:

1. the suprathermal He2+ fraction –> the ratio of He2+ to H densities with suprathermal energies

2. High energy field aligned beams near the foreshock edges show significant He/H ratios, whereas lower energy beams deeper within the foreshock exhibit intermediate proton distributions and lower He/H ratios

3. Still deeper (?) within the quasi-parallel shock, He distributions are nongyrotropic partial rings whereas H distributions are ring beams and density ratios return to solar wind levels.

4. Diffuse ions are found WHERE?. The ratio of suprathermal He to suprathermal H ion densities is similar to that for the solar wind composition.
Lines 100-102. The authors chose to simulate very rare solar wind conditions. There were only 85 hours of solar wind velocity between 700 and 800 km/s and densities less than 3 cm^-3 during the 17250 hours in the two-year period of 2012 and 2013 (0.5% of all conditions). Could the authors please add a paragraph to the conclusion stating what they expect the results for more typical solar wind conditions to be?

Line 150 says the simulation finds Nalpha/Nproton > 2 deep in the foreshock. Is that consistent with the summary above? What is the explanation for it? If the paper tells this somewhere and I have missed it, please strengthen the discussion to make it clear. I would have guessed that deep within the foreshock is a region of diffuse ions and I have read above that density ratios for diffuse ions are similar to those in the solar wind, not twice as great.

Line 181-182. When the authors present two case studies of observations they should tell where the spacecraft were located and present a plot showing the locations of the magnetopause and bow shock, the IMF lines, and the locations of the spacecraft. This will help in the comparisons and in the reader's comprehension.

Line 182. Actually it is probably the foreshock moving past the spacecraft and not vice-versa and the authors should make this clear.

In general (1). Where are the spontaneous hot flow anomalies reported and simulated to occur within the quasi-parallel foreshock? [Zhang et al., JGR, 118, 3357, 2013; Omidi et al; JGR, 119, 9823, 2014]

In general (2) Do the authors find foreshock compressional boundaries with density and magnetic field strength enhancements like those reported by Omidi et al. [JGR, 118, 823, 2013]? If so, where do these boundaries lie compared to those for the patterns for waves and suprathermal composition ratios?

I caught a few typos/corrections.

1. Author list. Stephen 2. Line 15. The –> to 3. Wilson III –> Wilson 4. Line 36

dynamical –> dynamic
* * *

---

## Author Comment (AC1) · 4 Aug 2020

We thank Dr. Sibeck for his helpful comments in improving our manuscript.

**This paper presents a comprehensive analysis of novel results from the Vlasiator model for proton and helium acceleration in the Earth's foreshock. The paper is clear and (in general) well written, the conclusions are substantiated by discussion of simulation results and comparison with observations. For the community interested in shock physics, the paper will be very important. For the general space plasma physicist., the results will be fairly important. Some of the results reported in the paper include (1) different edge locations for the proton and helium foreshock, (2) the manner in which the ratio of helium to proton**

[Figure]

density varies with location in the foreshock, (3) the nature of proton and helium distribution functions as a function of location, (4) the manner by which helium is heated in the foreshock, (5) the effects of the IMF orientation on foreshock boundary structure, and (6) the nature of waves/turbulence in and around the foreshock. I only have a few comments/questions.

Thank you for the helpful assessment of our key points and the significance of our study.

**Lines 50-61. I did not find this review of past work as clear as it could be. I have no objection to each sentence but I think it can be presented more carefully. The authors could systematically go through each region of space, or each type of distribution function, showing they are covering all possibilities. A table noting regions, types of distribution functions, and composition ratios would help. Could the authors just tell what is seen first, and then give explanations? Or could they state expectations and then tell what past work has seen? It would be cleaner than the mixture of observations and interpretations. Having a table would also be something useful that the authors could refer back to when summarizing their work towards the end of the paper, especially if they can check off each observation and state that their model predicted it.**

Thank you for the suggestion, a table might indeed help. We also noted the difficulty of comparing previously published $He^{2+}$ observations from different parts of the foreshock. A table was previously not included in order to not veer into review territory, but we will restructure this section and, if at all feasible, introduce a table and reference it in latter parts of the manuscript.

**Change:**
**1. the suprathermal $He^{2+}$ fraction $\rightarrow$ the ratio of $He^{2+}$ to H densities with suprathermal energies**

Thank you, that is a good formulation.

**2. High energy field aligned beams near the foreshock edges show significant He/H ratios, whereas lower energy beams deeper within the foreshock exhibit intermediate proton distributions and lower He/H ratios**

Agreed.

**3. Still deeper (?) within the quasi-parallel shock, He distributions are nongyrotropic partial rings whereas H distributions are ring beams and density ratios return to solar wind levels.**

Agreed.

**4. Diffuse ions are found WHERE?. The ratio of suprathermal He to suprathermal H ion densities is similar to that for the solar wind composition.**

Indeed, the typical location of diffuse ion populations (throughout the deep foreshock) should be mentioned.

**Lines 100-102. The authors chose to simulate very rare solar wind conditions. There were only 85 hours of solar wind velocity between 700 and 800 km/s and densities less than 3 cm$^{-3}$ during the 17250 hours in the two-year period of 2012 and 2013 (0.5% of all conditions). Could the authors please add a paragraph to the conclusion stating what they expect the results for more typical solar wind conditions to be?**

We are happy to add discussion regarding the simulation parameters. Despite the large speed and low density, the plasma beta (0.7) and Alfvenic Mach number (7) of the simulation are descriptive of quite typical solar wind conditions, and thus we do not expect these results to be atypical. A fast solar wind speed ensures efficient initialization of our simulation, which is computationally expensive, and thus allows for a longer simulated extent of well-formed foreshock dynamics.

**Line 150 says the simulation finds $N_{\mathbf{alpha}}/N_{\mathbf{proton}} > 2$ deep in the foreshock. Is that consistent with the summary above? What is the explanation for it? If the**

**paper tells this somewhere and I have missed it, please strengthen the discussion to make it clear. I would have guessed that deep within the foreshock is a region of diffuse ions and I have read above that density ratios for diffuse ions are similar to those in the solar wind, not twice as great.**

We believe this to be a feature of helium existing in more distinct structures (partial rings and gyrating clumps) than protons in the deep foreshock (as expected based on observations), and the fact that the diffuse population is removed from our simulation domain. This is mentioned section 4, but we shall add an initial note of this feature at this point.

**Line 181-182. When the authors present two case studies of observations they should tell where the spacecraft were located and present a plot showing the locations of the magnetopause and bow shock, the IMF lines, and the locations of the spacecraft. This will help in the comparisons and in the reader's comprehension.**

This is indeed a good suggestion, we shall add the plots of spacecraft locations and environmental conditions to Figures 3 and 4. Positions were already listed in section 3.1 along with verbal descriptions of the IMF, but sometimes a picture does say more than a thousand words.

**Line 182. Actually it is probably the foreshock moving past the spacecraft and not vice-versa and the authors should make this clear.**

We shall amend the wording to clarify the possibilities here. We did attempt to find crossings where the foreshock edge movement would be as slow as possible (no IMF discontinuities), so that it would be at least partially spacecraft movement instead of the edge sweeping over the spacecraft, but likely both effects are at play.

**In general (1). Where are the spontaneous hot flow anomalies reported and simulated to occur within the quasi-parallel foreshock? [Zhang et al., JGR, 118, 3357,**

2013; Omidi et al; JGR, 119, 9823, 2014]

SHFAs have been previously investigated within Vlasiator in Blanco-Cano et al. (2018) and a statistical study is under preparation (presented at EGU 2020, Tarvus et al.). A full study of helium dynamics in response to SHFAs would warrant a whole study of its own. Within the scope of this current study, we can state the following. As SHFA are identified via a flow deflection and an abundance of hot ions, we would expect to see SHFA only where there are plenty of beam-like, gyrating or intermediate suprathermal ions (diffuse ions will not significantly influence flow deflection). Our simulation shows that SHFAs are formed very close to the bow shock and deep within the foreshock, and the ratio of $He^{2+}$ to H densities with suprathermal energies is greater in this region than in the solar wind, likely due to the abundant diffuse proton population having been excluded. However, visual inspection of a number of SHFA-flagged regions shows that these do not show the suprathermal ion ratio rising much beyond 2. This is in agreement with an abundance of energetic non-diffuse protons within SHFA. Improvements to the numerical method or better analysis approaches may indeed merit a further study of alpha-particles within SHFAs, but we do not wish to investigate it in any more detail in this study.

**In general (2) Do the authors find foreshock compressional boundaries with density and magnetic field strength enhancements like those reported by Omidi et al. [JGR, 118,823, 2013]? If so, where do these boundaries lie compared to those for the patterns for waves and suprathermal composition ratios?**

The presence of foreshock compressional boundaries and their dependence on shock Mach numbers in Vlasiator simulations were investigated in Turc et al. (2018). As we do not expect them to be significant from an alpha-particle point of view, we have not discussed them in the manuscript, but can comment the following: Brief visual analysis of foreshock wave compressionality in the presented simulation indicates waves are compressional in the regions where Figure 1a shows well-structured (red-and-blue-striped) wave fronts. The positions at the bow shock where these two regions connect

may in fact be associated with the theorized "optimal" $\theta_{Bn}$-connection, but the compli-
cated wave interactions further in the flanks of the foreshock make this hard to discern.
We would also like to point out that especially in the region $X < 0$ at the flank we see
strong compressional features aligned with the IMF, akin to the canyons and ridges
which were seen in Blanco-Cano et al. (2018) and which were associated with caviton
and SHFA-type structures. These IMF-aligned ridges and canyons would, if advecting
past a spacecraft, appear like single structures such as cavitons and SHFAs.

**I caught a few typos/corrections.**
**1. Author list. Stephen**
**2. Line 15. The → to**
**3. Wilson III → Wilson**
**4. Line 36 dynamical → dynamic**

Thank you! These will be corrected.

---

## Author Comment (AC2) · 4 Aug 2020

Response to referee comment 1 by Jeffrey Broll

**The submitted work investigates the helium foreshock in simulation and MMS/HPCA observation. (Throughout, all mentioned helium is doubly ionized.)**

We wish to thank Dr. Broll for his helpful assessment of our manuscript.

**The main simulation of interest is 2D3V hybrid-Vlasov ions and massless-fluid electrons, with setup very similar to previously published work - low-$\beta$ fast solar wind, $45°$ cone-angle, supercritical. This is as appropriate for the study as is feasible, but I must wonder what acts in place of electron pressure gradients for the cross-shock potential. I suspect/guess (cf. I think Gosling+ 82, "Evidence for**
**spec. refl. ions upstr. from the $Q_\parallel$ bow shock", and Gosling + 89, "Ion refl. and downstream thermalization at the $Q_\parallel$ bow shock", it's been a while) that some deviation from the specular reflection ovals should be expected if the main shock ramp jump is smaller or less sharp, in particular - this is possibly a shortcoming of the reviewer, but some demonstration of the cross-shock potential at an interesting point or two would have been nice to see...**

We agree with the referee that a cross-shock potential field exists at astrophysical shocks, and that it has an effect on ion reflection, in particular at the quasi-perpendicular shock (it has been studied very little at the quasi-parallel shock front). The magnitude of the cross-shock potential, discerned from observations of quasi-perpendicular shock crossings, is estimated at 10% to 30% of the shock ram energy (Schwartz 1988). In the Ohm's law sense, the cross-shock electric field arises from a combination of the Hall current, electron pressure gradients, and drag due to the small population of gyrating ions at the shock front (Bale 2008). The majority ($\sim$ 60-70%) of the observed macro (i.e. ion)-scale electric field in the Normal Incidence frame is due to the Hall term $J \times B/ne$ (Eastwood 2007), which we already include in our model. Yang (2009) state that at the shock ramp, the shock-normal electric field is dominated by the ion Lorentz term and the Hall term, with the electron pressure term of negligible importance. Thus, hybrid models which neglect the electron pressure term do successfully model the majority of the cross-shock electric field, and only neglect one small portion of it.

Nevertheless, the role of cross-shock potentials on ion reflection is an interesting topic of study, and a study into different parametrizations (e.g. adiabatic or isothermal electrons) in a Vlasiator simulation is already underway (but not ready for submission yet). We will add discussion regarding this term to the manuscript, in particular to the section discussing specular reflection ovals.

**The data used are appropriate for the study (although more could have been gotten out of burst HPCA data as in the cited HPCA papers).**

We agree that burst HPCA data might be of great interest for many foreshock studies and thank the referee for the suggestion. We however fear that the low amount of helium ions detected by the instrument would likely require rebinning of the data in order to get statistically relevant results. Also, our survey of available burst mode intervals did not come across any foreshock crossings without an IMF discontinuity (a requirement which limited our options significantly).

**However it's not clear if the figures couldn't be better labeled - Figures 3's caption is traveling backwards in time.**

Could the referee please clarify this issue? We reviewed the figure and could not locate a problem, though of course would be happy to fix any errors.

**Further, the suprathermal proton and helium population seem to go above HPCA's energy range on 30 Dec 2018 so perhaps some kind of either Maxwellian or $\kappa$ fit to averaged foreshock E/Z slice would increase confidence that the suprathermals were being counted far enough up.**

Thank you for pointing out this issue regarding suprathermal ion energy ranges. A kappa-fit might indeed give some extra information regarding this, but the fine time detail, the low amount of energy bins, and the extent of statistical noise leads us to not taking this approach. Further, as the color bar shows, the phase-space density range is highly logarithmic, and densities are strongly dominated by the lowest included suprathermal energy bin(s). We shall add discussion to this effect to the manuscript.

**This issue and the different suprathermal definitions (which could be explained in the text for those not fluent in Vlasiator nuances) seem to make comparison harder than necessary.**

We admit that the difference in suprathermal definitions is unfortunate. The data amounts generated by Vlasiator necessitate reduction of data stored to disk and full VDF information is available only at select locations (as shown later in the manuscript).

MMS HPCA data could probably be converted into a Vlasiator-style suprathermal densities but that would not allow energy-time-spectrogram plotting without accurate estimation of GSE-coordinate motion of the spacecraft, and would result in a presentation method unlike those used in other spacecraft studies. We performed visual evaluation of HPCA distribution functions using the available binning and concluded that the energy threshold method used here is not wholly unreasonable. We will, however, amend the text to draw attention to this discrepancy and that some particles which would be likely included in the Vlasiator-style suprathermal population now fall below the energy threshold as seen by the spectrogram energy bin just below the threshold (and also at energies $<$100eV).

**The analysis lost me a bit right around the very large $T_\parallel/T_\perp$ - could the resolution be hindering something like firehose growth that would bring that down? This could be an issue; checking Gary's microinstabilities book (7.2.1 and fig. 7.1) and assuming $\beta$ sane, I'd guess that you could be missing some physics there. I think that given the nature of the work this is to be expected, but it could be mentioned for completion's sake.**

Thank you for an interesting point. Within the foreshock, the parallel plasma beta rises from the solar wind value of 0.7 to values ranging from 1 to 10 or even somewhat beyond in the field-aligned beam portion at the edge of the foreshock. At the same time, temperature anisotropies $T_\perp/T_\parallel$ decrease to as little as 0.3. This parameter range indeed begins to be firehose instable. However, the beam particles streaming along field lines away from the shock are (along with the frozen-in magnetic field) also efficiently convected laterally due to ExB drifts and enter the deeper foreshock region, where plasma parameters are in the stable region. One may theorize that non-uniform driving (e.g. including statistical noise) with a good spatial resolution might drive firehose microinstabilities right at the foreshock edge, and we are happy to add discussion to this effect to the manuscript.

**Having picked those nits I think this work should be published after some minor-**

**to-moderate complaints are addressed. In general there are some Vlasiator-specific nuances that are not common enough knowledge (at least for this reviewer) and possibly could be made so without detracting from the simulations section.**

Thank you for the kind suggestions. We shall strive to improve the clarity of the manuscript in these respects.

---

## Author Response (AR1)

Dear Editor,

Thank you for accepting our article into Annales Geophysicae. Although the acceptance was "as-is", we have included minor adjustments based on the helpful referee feedback, in accordance with the responses posted in the public discussion. We note that we refrained from performing a large restructuring of the introduction, instead focusing on the explicit clarifications suggested by David Sibeck. We also updated figures 3 and 4 with MMS locations as per his suggestion.

Best regards,

The authors